# Fungicides, herbicides and bees: A systematic review of existing research and methods

**Merissa G. Cullen**[1]☯, **Linzi J. Thompson**[2,3]☯, **James. C. Carolan**[1], **Jane C. Stout**[4], **Dara A. Stanley**[2,3]*

**1** Department of Biology, Maynooth University, Maynooth, Co. Kildare, Ireland, **2** School of Agriculture and Food Science, University College Dublin, Belfield, Dublin, Ireland, **3** Earth Institute, University College Dublin, Belfield, Dublin, Ireland, **4** School of Natural Sciences, Trinity College Dublin, Dublin, Ireland

☯ These authors contributed equally to this work.
* dara.stanley@ucd.ie

**Data Availability Statement:** All relevant data are within the manuscript and its Supporting Information files.

**Funding:** This research was funded as part of the PROTECTS project by the Irish Government's

## Abstract

Bees and the pollination services they deliver are beneficial to both food crop production, and for reproduction of many wild plant species. Bee decline has stimulated widespread interest in assessing hazards and risks to bees from the environment in which they live. While there is increasing knowledge on how the use of broad-spectrum insecticides in agricultural systems may impact bees, little is known about effects of other pesticides (or plant protection products; PPPs) such as herbicides and fungicides, which are used more widely than insecticides at a global scale. We adopted a systematic approach to review existing research on the potential impacts of fungicides and herbicides on bees, with the aim of identifying research approaches and determining knowledge gaps. While acknowledging that herbicide use can affect forage availability for bees, this review focussed on the potential impacts these compounds could have directly on bees themselves. We found that most studies have been carried out in Europe and the USA, and investigated effects on honeybees. Furthermore, certain effects, such as those on mortality, are well represented in the literature in comparison to others, such as sub-lethal effects. More studies have been carried out in the lab than in the field, and the impacts of oral exposure to herbicides and fungicides have been investigated more frequently than contact exposure. We suggest a number of areas for further research to improve the knowledge base on potential effects. This will allow better assessment of risks to bees from herbicides and fungicides, which is important to inform future management decisions around the sustainable use of PPPs.

## Introduction

Bees and other animal pollinators are beneficial in both the production of the majority of global crops and reproduction of the majority of flowering plant species [1, 2], with their value to global food crops estimated at €153 billion per year [3]. Declines in bees and other pollinators have been recorded over large geographic areas (e.g. [4–6]), which has led to concerns over sustainable food supply and the health of natural ecosystems [7]. While there are a

Department of Agriculture, Food and the Marine's Competitive Research Funding Programme (Grant Award No.17/S/232). The funders had no role in study design, data collection and analysis, decision to publish, or preparation of the manuscript.

**Competing interests:** The authors have declared that no competing interests exist.

number of contributing factors to pollinator decline, pesticide use (particularly in the form of Plant Protection Products (PPPs) in agriculture) has been highlighted as one of the key contributing drivers [8–10].

Although PPPs (which include insecticides, herbicides and fungicides) have many benefits for agriculture [11], there are also a number of potential risks associated with their use. These include aspects such as their role as a driver of pest resistance, resurgence and secondary pest outbreaks, as well as wider environmental contamination and human health concerns [12–15]. Although insecticides are applied to target insect pests, their use in agriculture can have implications for non-target insects that may be providing beneficial services to agriculture, such as pest control or pollination. While a range of beneficial insects may be exposed to insecticides, the majority of recent focus has been on social bees (primarily *Apis* and *Bombus*) at a range of different levels, with a particular focus on neonicotinoid insecticides and their lethal and sublethal effects at the molecular and cellular through to colony and population levels. Much of this work has been summarised in a number of key review papers (e.g. [16–22]).

Although insecticides are designed to directly affect insects, a range of other PPPs are used in modern agriculture in places where bees and other pollinators are active. These include fungicides, which target fungal disease, and herbicides, which target unwanted plants. Although insecticides are used globally, they are outweighed in terms of both tonnes of sales and tonnage applied [23, 24], and market value [25], by fungicides and herbicides. Much research on the environmental consequences of herbicide use has focussed on factors such as weed resistance due to over-use [26, 27] and water and soil contamination [15, 28–30], whilst both fungicides and herbicides have been scrutinized for possible effects on human health [14, 31, 32].

As herbicides and fungicides are not designed to target insects, little is known as to whether they pose a risk to bees and other insect pollinators. However, emerging evidence has suggested that herbicides can affect factors such as bee navigation, learning and larval development [33–35] whereas fungicides can affect food consumption, metabolism and the immune response [36–38]. Bees may be exposed to these compounds directly via contact exposure during or after application, or via oral exposure through contaminated nectar and pollen. In order to minimise impacts on non-target pollinating insects such as bees, it is important to understand any potential effects these compounds may have, to determine the risks they pose and to mitigate against them.

Here, we review the current state of knowledge of the impacts of fungicides and herbicides on bees by means of a systematic literature review. Specifically, we set out to answer the following questions:

1. In what year and in what geographical location has existing research taken place?

2. Which bee species have been most commonly studied?

3. How have bees been experimentally exposed to PPPs?

4. What methodological approaches have been used?

5. Which fungicides and herbicides have been most commonly studied, and have studies used analytical grade active ingredient and/or PPPs as part of a commercially produced pesticide formulation?

6. Do studies claim to use field realistic doses of the compounds used?

7. What life stages, effect levels and types of effect have been investigated (see Table 1 for further explanation)?

**Table 1. The information extracted from each paper as part of the systematic review.**

| Variable | Levels |
|---|---|
| General information | Bibliographical reference, country, bee species and family studied |
| Exposure | · Topical–bees treated with PPPs via direct contact |
| | · Internal–via injection to organs in situ or in vitro |
| | · Oral—through nectar/sugar solution, including where larvae were fed a PPP-treated diet consisting of royal jelly or other commercially-available substances aimed at larval nutrition. |
| | · Oral—through pollen |
| | · Oral—nectar/sugar solution and pollen |
| Methodological Approach† | · Laboratory |
| | · Semi-field–outdoors but confined to e.g. exclusion cages |
| | · Field–outdoors with no restriction on bee movement |
| | · Model |
| | · Combined |
| Pesticide Type | · Substance group |
| | · Fungicide |
| | · Herbicide |
| | · Fungicide and Herbicide |
| | · Fungicide and/or herbicide combined with any other PPP(s) (combined studies) |
| | · Analytical grade |
| | · Formulation |
| | · Analytical grade and formulation |
| Field Realism | · Author claims field-realism of study |
| | · Author does not explicitly claim field-realism of study |
| Life stage* | · Egg |
| | · Larvae |
| | · Pupae |
| | · Adult |
| Effect Level | · Population–effect on a measured population of bees |
| | · Colony–effect measured on a bee colony e.g. reproduction, biomass, survival |
| | · Individual–effect measured on an individual bee e.g. mortality, behaviour |
| | · Sub-individual–effect measured within an individual bee e.g. genomics studies |
| Effect type | · Foraging Ability |
| | · Nesting |
| | · Learning ability |
| | · Other behaviour |
| | · Male production |
| | · Queen production |
| | · Biomass |
| | · Vulnerability to other stressors |
| | · Pollination services |
| | · Genomic |
| | · Physiological function and morphology |
| | · Sensory (e.g. gustatory or olfactory) |
| | · Consumption |
| | · Mortality |
| | · Navigation |
| | · Other |

† as per Lundin et al. [16].

*Life stage was recorded as the stage at which bees were exposed to PPPs, and the stage that effect types were measured on. If these were different then both were included in final analyses.

By quantifying the number of studies that have investigated the direct impacts of herbicides and fungicides on bees, we can determine the current state of knowledge surrounding this subject area and identify gaps for future research to inform risk assessments and sustainable PPP use.

## Methods

We used the Web of Science Core Collection to search for available literature on fungicides, herbicides and bees. The database was accessed on 09 November 2018, and queried using the following search terms to capture relevant literature: (fungicide*) AND (*bee OR *bees) and (herbicide*) AND (*bee OR *bees). This resulted in a primary dataset of 437 publications. To ensure consistency of the research included in our review we chose to use only peer-reviewed journal articles which reported primary research, and therefore non-peer reviewed articles were removed from the dataset. Duplicates (37) and non-accessible peer-reviewed articles (16) were removed (see Prisma table and flow diagram in S1 Fig & S1 Table), resulting in 385 papers remaining. These were then further examined to determine if our review criteria were met, namely that the paper must have reported an investigation of the direct effects of at least one fungicide and/or herbicide on any bee species. The types of effects reported included investigations into (i) toxicity/mortality effects, (ii) sublethal behavioural effects, including foraging ability, (iii) susceptibility to other stressors e.g. disease or combinations with other compounds, (iv) impacts on a colony or population level, (v) impacts on bees at a genetic, molecular, or physiological level, (vi) impacts on pollination services delivered by bees (only where bees had been directly exposed to the fungicide/ herbicide), or (vii) a modelling approach focusing on the direct impacts of herbicides and fungicides to bees. Studies that examined indirect herbicide and fungicide effects e.g. effects on nectar quality or microbiota, or that used observational rather than experimental methods e.g. observed changes to bee communities as a result of PPP spraying, were omitted from our analysis as patterns may not have been directly due to use of PPPs. The final dataset included 89 papers (S2 Table).

For each paper, we extracted the following information: full bibliographical reference, country, bee species, exposure method, methodological approach, PPP product name (where relevant), active ingredient and PPP substance group, whether the author claimed a field realistic dose and the effect level and type studied (Table 1, S2 Table). Where multiple categories of any variable were reported in the same paper, all were included in final analyses.

Exposure methods were first classified as to whether bees were treated with PPPs via contact exposure ('topical exposure'), or 'internal exposure' where PPPs were introduced to internal organs via injection, or to organs previously removed from bees. Where bees were fed PPPs orally ('oral exposure'), the medium through which PPPs were delivered was categorized as 'oral through nectar/sugar solution', 'oral through pollen', or 'oral through both pollen and nectar/sugar solution'. Larval studies were classified as exposure via 'oral through nectar/sugar solution' where larvae were fed a PPP-treated diet consisting of royal jelly or other commercially-available substances aimed at larval nutrition.

Methodological approaches were divided into five categories: 'laboratory', 'semi-field', 'field', 'model' and 'combined' as per Lundin et al. [16]. 'Laboratory' studies were defined as those carried out within the laboratory. 'Semi-field' studies were defined as those that were conducted outdoors, but confined bees, e.g. using exclusion cages. 'Field' studies were defined as studies conducted outdoors with no restriction on the bees' movements and data were collected in the field. "Model" studies were any studies that used a modelling approach to determine outcomes rather than an experimental approach. 'Combined' studies were defined as studies which used multiple methods for the same endpoint.

PPP type (fungicide, herbicide or both) was also recorded from each study. Studies were also analysed to determine whether compounds were studied as formulations or analytical grade standards. Where studies did not distinguish between formulation or analytical grade standard PPP use, we assumed they were analytical grade, as formulations are normally reported given they may contain other chemicals (e.g. surfactants).

To elucidate what herbicide and fungicide groups have been most widely studied, we also recorded the substance group/class of each PPP, informed by the University of Hertfordshire Pesticide Properties Database [39] or equivalent. Studies that investigated the impact of multiple PPP exposure on bees at any level were classified as 'combined' due to difficulties in the clarification of, and differences between, synergistic, additive and potentiation effects of PPPs. Field realism was defined simply on the basis of whether authors claimed to have investigated herbicides and/or fungicides at a field-realistic concentration. If studies did not explicitly state this, it was assumed that the study was not carried out using field-realistic concentrations of PPPs.

The level of biological organisation at which each study was conducted was defined as "effect level", differentiated by four categories: 'Population', 'Colony', 'Individual', and 'Sub-individual' (following the approach of Lundin et al. [16]). 'Population' was defined as an effect on a measured population of bees; 'colony' as any effect on the colony as a whole, such as reproduction, biomass, growth or survival; and 'Individual' was defined as any impact to the individual such as mortality or sub-lethal effects e.g. behaviour and learning. 'Sub-individual' was defined as any impact within the individual, including impacts on a genomic or physiological level. Information was also extracted on the life stage that individual and sub-individual level studies focused on (adult, pupa, larva or egg). If the life stages at which bees were exposed to pesticides differed to the life stage at which effects were measured, then both were included in final analyses.

Based on our knowledge of the existing literature and previous work [40], we described fifteen different "effect types" that were assessed including: foraging ability, nesting, learning ability, other behaviour, male production, queen production, biomass, vulnerability to other stressors, pollination services, genomic, physiological function and morphology, sensory (e.g. gustatory or olfactory), consumption (e.g. ability to consume nectar and/or pollen), mortality or navigation. For each study, we attributed the research to one or more effect types, and included an 'other' column for any studies that did not fit into these categories. Studies were placed into multiple categories if they contained more than one effect type.

Where studies included more than one option in any of the variables measured, it was included in analyses of both. For example, if a study included results of effects on both pupal and adult life stages, it was counted in both these categories in the results. Finally, a brief descriptive summary of the main findings of each study included in our systematic review was determined from information provided in the abstract of each paper, and collated for qualitative purposes (see S3 Table).

## Results

### When and where did studies take place?

Although the first of the 89 papers was published in 1973, there was at most one study per year until 1996, with an exponential increase in the number of studies published more recently (e.g. since 2010; Fig 1). Most studies were carried out in North America (43 studies) and Europe & Russia (29 studies) followed by South America (8 studies), Asia & Middle East (6 studies), Africa (1 study) and Oceania (2 studies; Fig 2).

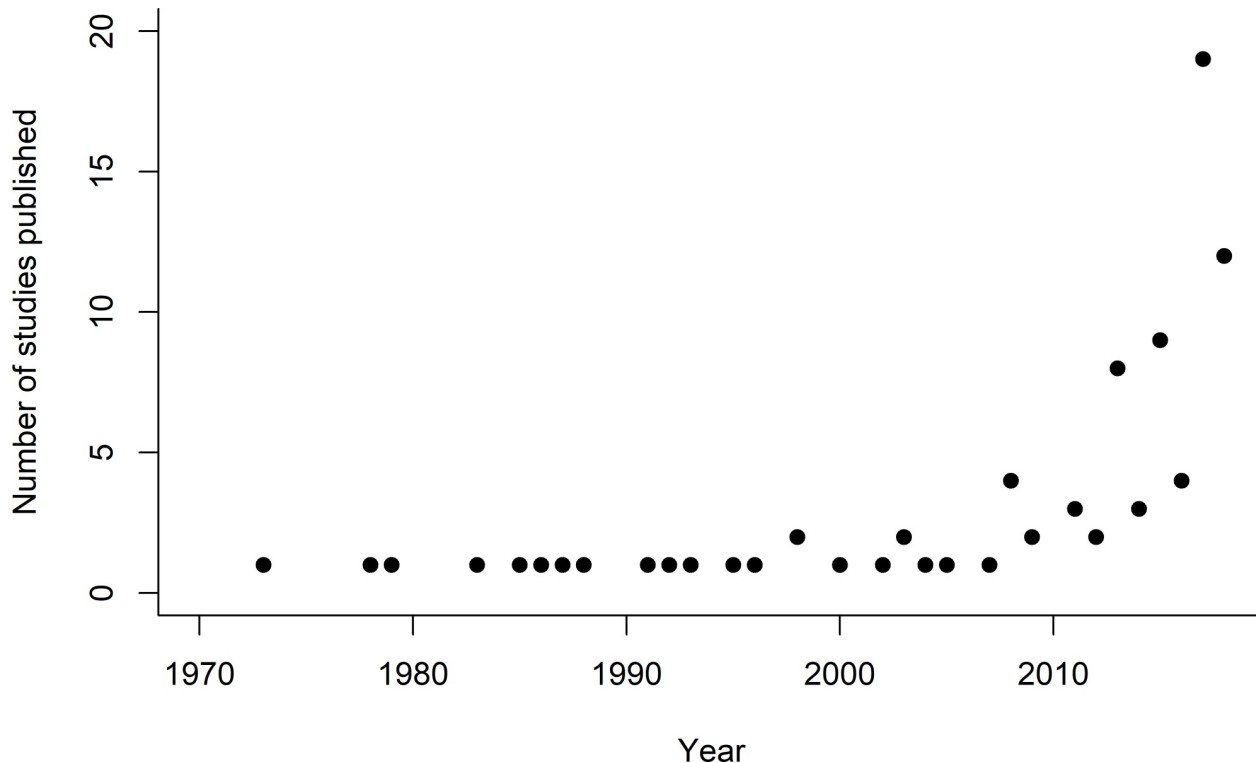

**Fig 1. The number of studies on herbicides and/or fungicides and bees that met the criteria for inclusion in this review, and the year they were published (one paper that appeared in our search but that was scheduled for publication in 2019 is not included).**

### Which bee species have been most widely studied?

The vast majority of studies investigated herbicide and fungicide effects on *Apis* species (67 studies, Table 2); of which the majority focused on *Apis mellifera*, with two studies on *Apis cerana*. A smaller number of studies addressed other bee species, including *Bombus* spp. (13 studies), *Osmia* spp. (7 studies), *Megachile rotundata* (8 studies) and other bee species (4 studies). Within the bumblebees, *Bombus terrestris* was most widely studied (8 studies), followed by *Bombus impatiens* (4 studies), with only one study investigating a variety of other bumblebee species (*B. occidentalis*, *B. affinis*, *B. pensylvanicus* and *B. terricola*). *Osmia* species included *Osmia bicornis* (3 studies), *O. lignaria* (3 studies) and *O. cornifrons* (1 study). Most studies investigated effects on just one species, and only nine examined effects on multiple species in the same paper. Nine bee species studied were social, whereas four species were solitary.

### How are bees experimentally exposed to PPPs?

Bees were predominantly exposed to PPPs orally (62 studies), through nectar (41 studies), pollen (11 studies), or both (11 studies). Thirty-five studies exposed bees topically. Five studies exposed bees internally, either by injection or by testing of isolated internal organs. Most studies used chronic exposure (57 studies), with fewer carrying out acute exposures (40 studies).

### What methodological approaches have been used?

The vast majority of studies were carried out in the lab (74), with nine carried out in semi-field conditions, and 12 at full field scale. Nine studies used a combination of different methodological approaches, and four studies used a modelling approach.

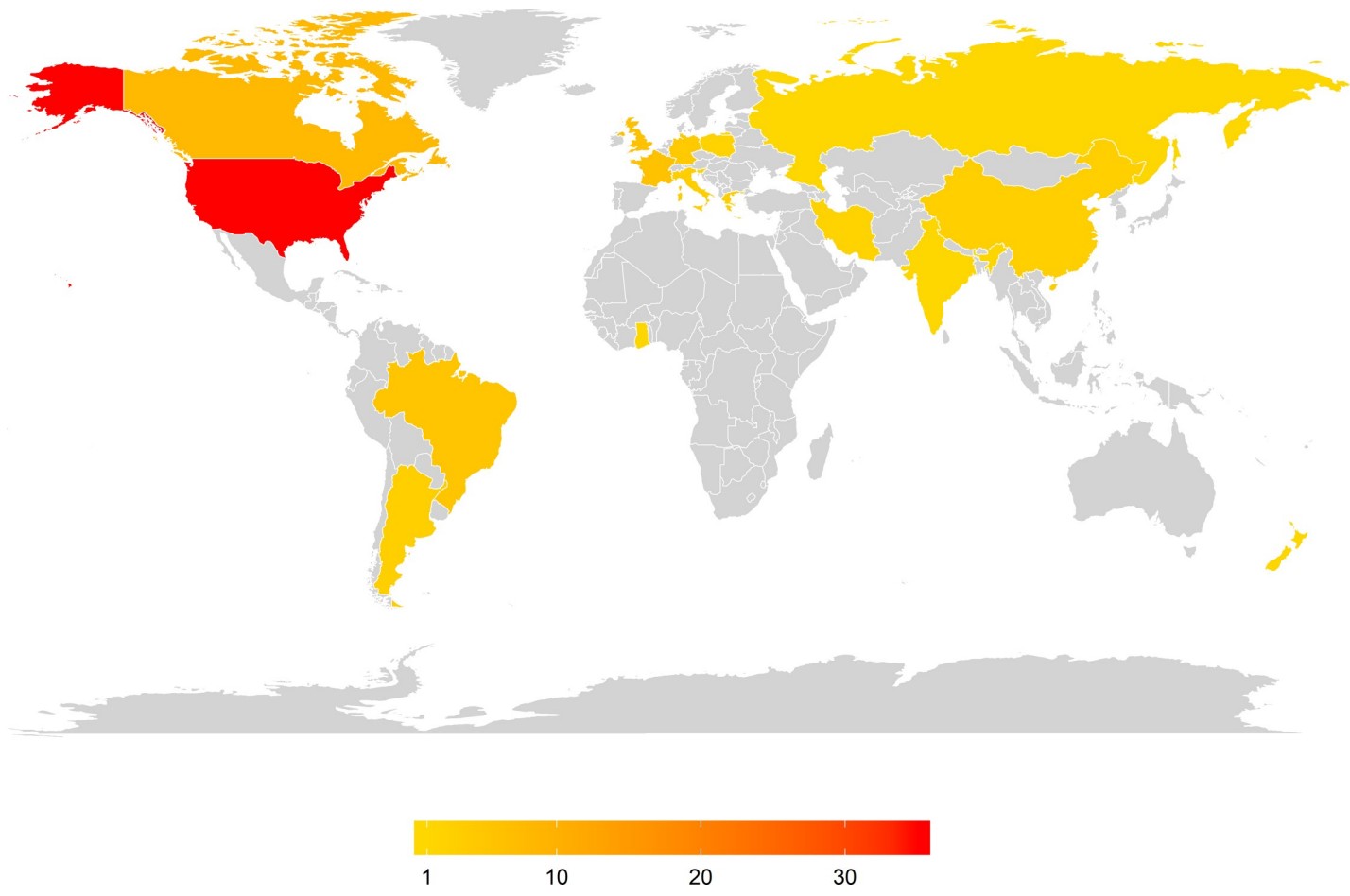

**Fig 2. The number of studies undertaken in each country worldwide (grey = 0 studies).** The majority of studies have been undertaken in North America and Europe, followed by South America.

**Table 2. The numbers of studies investigating the effects of herbicides and/or fungicides on different bee species.**

| Bee species | Number of studies | Social or solitary |
|---|---|---|
| *Apis mellifera* | 66 | Social |
| *Bombus terrestris* | 8 | Social |
| *Megachile rotundata* | 8 | Solitary |
| *Bombus impatiens* | 4 | Social |
| *Osmia bicornis* | 3 | Solitary |
| *Osmia lignaria* | 3 | Solitary |
| *Apis cerana* | 2 | Social |
| *Osmia cornifrons* | 1 | Solitary |
| *Other bumblebee species* | 1 | Social |
| *Partamona helleri* | 1 | Social |
| *Tetragonisca angustula* | 1 | Social |
| *Tetragonisca fiebrigi* | 1 | Social |
| *Hypotrigona ruspolii* | 1 | Social |

**Table 3. The most studied herbicides ranked in descending order, and whether they were studied individually or in combination with other PPPs.** The number of studies that used a formulation or the analytical grade active ingredient alone, are displayed in parenthesis.

| Herbicide | Pesticide group | No. studies (formulation, active ingredient) | Individual / Combination |
|---|---|---|---|
| 1. Glyphosate | Phosphonoglycine | 15 (4, 11) | 15 / 2 |
| 2. Atrazine | Triazine | 6 (2, 4) | 6 / 2 |
| 3. 2,4-D | Aryloxyalkanoic acid | 5 (1, 4) | 5 / 1 |
| 4. Paraquat | Bipyridylium | 5 (1, 4) | 5 / 0 |
| 5. Simazine | Triazine | 4 (0, 4) | 3 / 1 |

## Which fungicides and herbicides have been studied?

Most studies investigated the effects of fungicides (70 studies) compared with herbicides (29 studies). These effects were measured as a result of exposure to individual compounds or a combination of different compounds. Fifty-one and 14 papers reported studies on ≥2 fungicides or herbicides in the same study, respectively. The majority of studies report an investigation into compounds individually; however, of these studies 19 investigated the combined effects of fungicides and insecticides, one on fungicides and herbicides, and three on herbicides and insecticides.

**Herbicides.** The top herbicide substance groups studied were phosphonoglycine (15 studies) and triazine (8 studies) followed by alkylchlorophenoxy and bipyridylium (5 studies each; S4 Table). The most widely studied herbicide compound was glyphosate, which was investigated in 15 studies, followed by atrazine, 2,4-D, paraquat and simazine (Table 3). For these top compounds, most studies examined the impacts of herbicides alone while some also investigated combined effects with other compounds. Active ingredients were used more frequently than formulations (Table 2). The remaining 15 herbicides were investigated in ≤2 studies each, both as part of a formulation, and/or analytical grade active ingredients (S4 Table).

**Fungicides.** The top fungicide substance groups reported were triazole (30 studies), strobilurin (15 studies) and imidazole (14 studies), either alone or in combination with other PPPs or substances (S4 Table). In terms of individual compounds, propiconazole was the most widely investigated, followed by boscalid, chlorothalonil, pyraclostrobin, iprodione, prochloraz and myclobutanil (Table 4). These compounds varied in whether effects were investigated alone and/or in combination with other compounds, and both formulations and active ingredients only were used (Table 4). The remaining 53 fungicides were investigated in ≤6 studies using formulations and analytical grade chemical standards (S4 Table).

## Do studies claim field relevance?

Overall, 42 studies claimed to use field realistic concentrations of PPPs in their investigation. Two studies were not analysed for field realism as it was not relevant in the context of the research, e.g. treating bees with fungicide to control bee disease.

**Table 4. The most studied fungicides ranked in descending order, and whether they were studied individually or in combination with other PPPs.** The number of studies that used a formulation or analytical grade active ingredient alone, are displayed in parenthesis.

| Fungicide | Pesticide group | No. studies (in formulation, active ingredient only) | Alone / in combination with other compounds |
|---|---|---|---|
| 1. Propiconazole | Triazole | 15 (8, 7) | 11 / 8 |
| 2. Boscalid | Carboximide | 13 (9, 4) | 3 / 10 |
| 3. Chlorothalonil | Chloronitrile | 12 (3, 9) | 10 / 3 |
| 4. Pyraclostrobin | Strobilurin | 12 (9, 3) | 2 / 11 |
| 5. Iprodione | Dicarboximide | 11 (9, 2) | 10 / 3 |
| 6. Prochloraz | Imidazole | 11 (2, 9) | 10 / 7 |
| 7. Myclobutanil | Triazole | 8 (3, 5) | 6 / 2 |

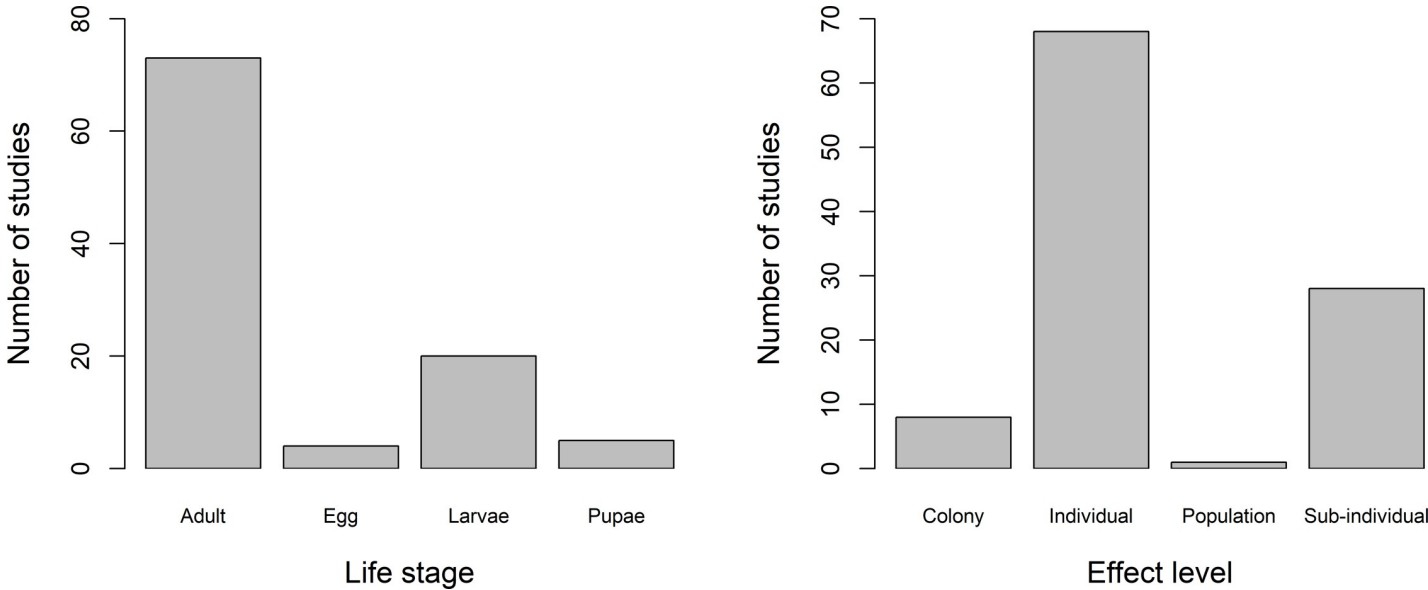

**Fig 3.** The number of studies found in this review on the impacts of herbicides and fungicide effects on bees at various life stages (left) and effect levels (right).

### Life stage

Most studies focussed on the effects of herbicides and fungicides on adult bees (73 studies across all species), followed by larvae (20 studies), pupae (5 studies) and the egg stage (4 studies; Fig 3). The pupal and egg stage were only included in studies which assessed effects across multiple life stages for studies at both the individual and sub-individual level. Egg studies were only carried out on *Megachile rotundata*, whereas larval and pupal studies involved multiple bee species.

### Effect level

The majority of studies focussed on the effects of fungicides and herbicides at the individual/ whole organism level (68 studies), followed by the sub-individual level (28 studies), colony level (8 studies) and population level (1 study; Fig 3).

The majority of studies investigated effects on adult bees at the individual level (73 studies), followed by effects on adult bees at the sub-individual level (23 studies). Few studies either assessed effects on bees across multiple life stages or exposed bees to fungicide/herbicide at earlier life stages and then assessed the effects at a later life stage; 29 studies at the individual level and 8 studies at the sub-individual level.

### Effect type

Most studies investigated multiple effects (58 studies) of fungicides and/or herbicides on bees but there was still a large focus on investigating one effect (31 studies). On average studies investigated two effect types. The most widely studied effect type was mortality, both for studies investigating a single effect type (20 studies) and multiple effect types (43 studies).

Effects of fungicides and/or herbicides on mortality (63 studies) and impacts at the physiological/ morphological level (27 studies) were the most widely studied areas, with all other effect types categorised investigated in ≤14 studies (Fig 4). A summary of the main findings of each paper is presented in S3 Table.

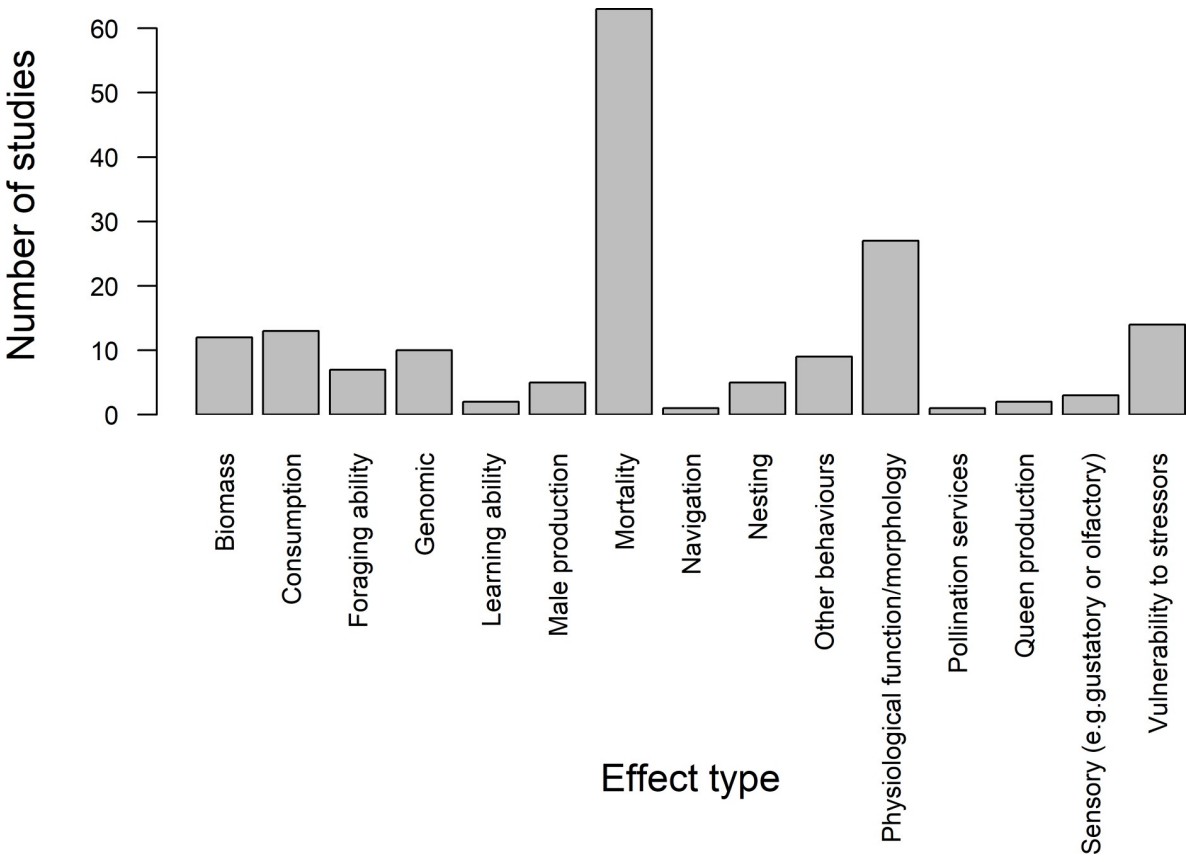

**Fig 4. The number of studies found that tested the impacts of fungicides and/or herbicides on each effect type.**

## Discussion

With a growing global population and increasing environmental concerns, it is crucial to sustainably manage our agricultural systems. It is important to understand both the benefits and risks of PPP use to humans and the environment in order to make decisions around agricultural management. Pesticide use has been linked with pollinator declines globally. Understanding what has been studied in terms of different pesticide classes and substance groups and potential impacts on bees and other insect pollinators is vital to determine the role of PPPs in bee decline.

Recent political, public and scientific interest in the non-target effects of insecticides on bees has out-weighed research into impacts of other PPPs. For instance, a recent review focusing on the impacts of a single class of insecticides, the neonicotinoids, on bees yielded 543 papers on an initial search [with 268 included in the review; 16], while in this review, focussing on all classes of fungicides and herbicides, we only found 437 papers (of which 90 were relevant to this review). Although herbicides and fungicides are not designed to target insects, the current declines in both diversity and abundance of some bee species [4–6] and attempts to resolve the factors driving these declines warrant the scrutiny on the potential effects of non-insecticide PPPs on pollinators.

The majority of studies on the effects of herbicides and fungicides on bees have been undertaken in North America and Europe, especially from the mid-2000s onwards; a trend similar to what is seen in the literature on neonicotinoids and bees [16]. However, PPPs are used

globally, and the compounds used and conditions they are used in may differ significantly in different regions of the world. Although PPPs are used most heavily in developed countries, they are increasingly being used in other parts of the world where regulations and best practice around their environmental impacts may not be as stringent [41]. In addition to this, most insect pollinated crops are grown in countries outside the EU and North America [3]. All these factors suggest that it is important to investigate the impacts of fungicides and herbicides on pollinators in the conditions they are used globally, to build a full picture of any potential impacts.

As with work on insecticides, we have found that the majority of studies investigating effects of fungicides and herbicides on bees have focused on the European honeybee (*Apis mellifera*) as a model species. While this species is an important pollinator worldwide [42], it is only one of approximately 20,000 species of bees globally [43]. There is evidence to suggest that bee species differ in terms of their responses to insecticides in both lab and field based studies [44–48], and it is likely that this may also be the case for other PPP classes. Our work shows that bumblebees and solitary bees are not as well represented in the literature as honeybees in terms of effects of herbicides and fungicides, and may be worthy of additional attention for a more holistic appraisal of the effects of PPPs on bees. Although the majority of bee species globally are solitary [43], we have demonstrated a bias towards research on social species. Social bees, due to their large colony sizes (especially *Apis mellifera*) may be able to buffer negative impacts of PPPs, leading to a higher resilience at the colony and population level. This potentially renders research on social bees inapplicable to solitary species [44, 45] which have differing biology and can have different potential exposure routes [49, 50]. In addition, we have shown that most research focuses on social bees in the context of impacts on individuals, but it is important to consider impacts at the colony level since this is the true reproductive unit.

The vast majority of papers we reviewed focused on the effects of herbicides and fungicides on bee mortality (71%). Traditionally, the risks of these compounds to bees were assessed using LD50 tests. However, there could be sub-lethal effects that this approach does not detect, as has been demonstrated with insecticides [19, 40]. The number of studies investigating potential sub-lethal effects of herbicides and fungicides on bees is low (29% focussed only on sublethal effects, 48% examined both lethal and sublethal effects), making this an area worthy of further attention.

For herbicides, we found that most studies examined the impacts of active ingredients rather than formulations, whereas for fungicides this was much more compound specific (Tables 3 & 4). Using active ingredients allows the investigation of the impacts of that compound alone. However, commercially, most PPPs are supplied as formulations and so this may be a more likely exposure route to bees in the environment. Formulations contain a variety of substances other than the active ingredients (e.g. adjuvants), and it could be that some of these other substances may also interact with bees. A number of different PPPs can be applied to any piece of land in close temporal proximity, and so there is a vast number of possible PPP combinations applied to land across the globe with differing persistence's in pollen, nectar and soil, and boundless possible interactions between them. Realistically, the number of possible PPP combinations bees could be exposed to is too large for any one study. However, it's important that the role of single compounds versus mixtures of compounds (alone and in formulation) in terms of effects on bees and other pollinators continues to be investigated in future studies.

For both herbicides and fungicides, there are a few compounds that have received most attention (Tables 3 & 4), with many chemicals available on the market that have not been investigated in the scientific literature in terms of their effects on bees. While it may not be possible to evaluate the effects of all compounds, there are certain considerations that could be

taken into account when determining which compounds may pose a higher risk to bees. For example, it is likely that compounds used on flowering crops where bees forage may be of higher risk than those used on non-flowering crops, and compounds that are water soluble and systemic are more likely to end up in nectar and pollen than those that are not (e.g. the neonicotinoids). In 2013, the European Food Safety Authority (EFSA, the European organisation that carries out food-related risk assessments and informs findings to the public) guidelines for the risk assessment of PPPs on bees were developed, along with increased understanding of different exposure routes and differing risk levels of PPPs on different bee species (European Food Safety Authority, 2013). Previous guidelines focused on mortality of adult honeybees as a result of acute oral exposure to PPPs. This new guidance document is a step forward for the rigorous testing and regulation of pesticide risk to bees. It suggests that risk assessment is carried out on *Apis mellifera*, *Bombus* spp. and solitary bees to gain a more comprehensive and clearer understanding of PPP effects on bees as a group and not just a single species. Additional assessment factors are also suggested for bumblebee and solitary bee species when extrapolating from honeybee endpoints, to account for taxon-specific responses. Furthermore, the guidelines propose that testing should be carried out for compounds (and their specific uses) that are more likely to be of risk to bumblebees and solitary bees. The document suggests a tiered risk assessment system, starting with a lower cost-effective tier and working its way up to a higher complex and likely more expensive tier if previous results call for such investigation. In addition to the new risk assessment suggesting testing for chronic exposure, different routes of exposure are also considered, for example contact exposure via dust and spray drift and oral exposure via contaminated water. However, testing under these guidelines is not always necessary and the risk assessor decides which exposure routes are likely after consideration of all possible exposure routes.

One way to direct future research into the effects of PPPs on bees is to consider if and/or how bees could come into contact which these compounds. Bees may come in contact with herbicides and fungicides in the environment in two main ways; by contact or topical exposure when the bee is directly sprayed or comes into physical contact with a sprayed plant part, or by oral exposure via foraging for nectar and pollen on a herbicide or fungicide-treated plant [51, 52]. We have shown that much of the existing literature has investigated the effects of oral exposure, while topical exposure has been less well studied. This may be due to difficulties in determining or mimicking field-realistic topical exposure in an experimental setup, or due to the assumption that oral exposure is the most likely route of exposure. Residues of both fungicides and herbicides have been found in the nectar and pollen of both crops and wild plants visited by bees [51, 52] although information is limited (Zioga et al. in prep) and we do not fully understand the relative risks of both and how they persist in the environment. Furthermore, fungicides and herbicides could pose a risk to pollinators due to their varying persistence in the environment, leading to multiple pesticide exposure even where PPPs were not applied in close temporal proximity. Metabolites of PPPs applied could also pose a risk to pollinators e.g. the commercially used neonicotinoid insecticide clothianidin is the metabolite of the commonly used neonicotinoid thiamethoxam. Another factor to consider is that exposure risks to bees may differ between agricultural settings where users are trained in PPP application, and amenity use where PPPs are available to members of the public. The likelihood of bees coming into contact with a particular compound, and whether or not that compound has already been studied extensively, should also be considered.

In this review, we have focused on the direct effects of fungicides and herbicides on bees. However, it is important to acknowledge that there may also be a suite of secondary and/or knock-on effects not covered here. For example, one of the likely largest impacts of herbicide use for pollinators is the reduction or removal of flowering plants to forage on, and many

studies have shown that organic farms have more available forage for pollinators [e.g. 53, 54]. There is also evidence to suggest that herbicide use could affect gut microbiota of bees with consequences for both nutrition and susceptibility to pathogens [54, 55], and there could be indirect effects on bees through soils as has been demonstrated for insecticides [56]. These secondary effects of PPP use on bees and their available forage should also be a focus of future research.

In order to ensure we searched high quality and peer-reviewed literature, we searched through all citation indexes available on Web of Science. While we are confident that the majority of peer-reviewed literature has been analysed and accurately summarised in this review, we could not determine if all relevant studies to date have been identified in our search. For example, not all journals meet the requirements for inclusion on Web of Science databases, and there can be a time lag between publication and appearing on the system. Although some non-peer reviewed industry literature may also be of relevance to our review, for consistency only peer-reviewed academic literature was included. As we used the search terms "herbicide" and "fungicide", it is likely that some studies may have only referred to compounds they investigated by their direct name rather than by their pesticide class; however due to the number of potential compounds within these classes, searching each one individually was not possible. In addition, older studies included in Web of Science may not have specific keywords in titles or searchable abstracts, and so these studies could have been absent from our search results. However, it is unlikely that inclusion of these studies would hugely alter our results due to the dramatic increase in publications in this area since approx. 2010; a trend observed in literature on other PPPs and their effects on bees [16].

While we have investigated the range of research approaches that have been used to study potential effects of herbicides and fungicides on bees and provided a summary of main findings (S3 Table), a full evaluation of what effects were found in these studies and their direction (e.g. a meta-analysis) was beyond the scope of this study. There are existing meta-analyses examining the effects of insecticides on various aspects of bees and their behaviour where there is a larger body of work assessing particular compounds, scenarios or behaviours [46, 57, 58]. As there are over 90 studies included in this review this shows that there is a building body of literature looking at the effects of fungicides and herbicides on bees. However, we have shown that existing studies evaluate wide range of compounds, bee species, exposure routes and concentrations of pesticide, and as such, there is not yet a sufficient body of literature to perform a meta-analysis of more specific and targeted questions. This review is a first step in addressing this issue, and when certain research gaps are filled this area may benefit from a meta-analyses in the future to build a picture of the magnitude and direction of any effects.

With an increasing interest in the sustainable use of PPPs, and public scrutiny over the non-target effects of herbicides and fungicides in a range of areas including human health, water contamination and pest resistance, it is likely that there will be more interest in the impacts of herbicides and fungicides on bees in the future with new research emerging over time. Neonicotinoids were first released in the 1990s and their use has increased exponentially in the last 20+ years. However, during this time, pollinators were exposed to a variety of these insecticides. After decades of scientific research into the negative effects of neonicotinoids on pollinators, the outdoor use of three commercially used neonicotinoids were banned in 2018 in the European Union. Therefore, it is important to avoid the prolonged and unsustainable use of PPPs if further pollinator declines are to be prevented. In order to ensure safe pesticide use, it is important to determine the effects of commercially used PPPs on multiple bee species using methodology that accurately reflects all possible exposure routes. Although there is an increasing amount of scientific literature regarding herbicide and fungicide effects on bees, several key knowledge gaps in our current understanding remain. These include a lack of

studies on bumblebee and solitary bee species, the low number of studies considering contact and internal exposures and the lack of attention to certain compounds over others. To fully understand the potential risks of herbicides and fungicides to bees and to mitigate against them more research is required, specifically diversifying the type of research (i.e. exposure route, study species and type of exposure) and the range of compounds investigated. It is important to address these gaps in the future if we are to build a body of research capable of contributing towards future policy and ensure the sustainable management of agricultural systems and continued provision of pollination services to both crops and wild plants.

## Supporting information

**S1 Fig. PRISMA flow diagram.**
(DOCX)

**S1 Table. PRISMA checklist.**
(DOC)

**S2 Table. Raw data extracted from each of the 89 papers.**
(XLSX)

**S3 Table. Summary of the main findings of each paper included within the review.**
(XLSX)

**S4 Table. The top fungicide and herbicide substance groups.**
(DOCX)

## Acknowledgments

We would like to thank the other members of the PROTECTS project research team (Blanaid White, Matt Saunders, Elena Zioga, Mathavan Vickneswaran, Michael Kitching and Aoife Delaney) for useful discussions and enthusiasm.

## Author Contributions

**Conceptualization:** Merissa G. Cullen, Linzi J. Thompson, James. C. Carolan, Jane C. Stout, Dara A. Stanley.

**Data curation:** Merissa G. Cullen, Linzi J. Thompson.

**Formal analysis:** Merissa G. Cullen, Linzi J. Thompson, Dara A. Stanley.

**Methodology:** Merissa G. Cullen, Linzi J. Thompson, Dara A. Stanley.

**Supervision:** James. C. Carolan, Jane C. Stout, Dara A. Stanley.

**Writing – original draft:** Merissa G. Cullen, Linzi J. Thompson, Dara A. Stanley.

**Writing – review & editing:** Merissa G. Cullen, Linzi J. Thompson, James. C. Carolan, Jane C. Stout, Dara A. Stanley.

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
