## [Decision Letter · Decision Letter 0]

13 Sep 2019

PONE-D-19-21233

Fungicides, Herbicides and bees: A systematic review of existing research and methods

PLOS ONE

Dear Dr Stanley,

Thank you for submitting your manuscript to PLOS ONE. After careful consideration, we feel that it has merit but does not fully meet PLOS ONE’s publication criteria as it currently stands. Therefore, we invite you to submit a revised version of the manuscript that addresses the points raised during the review process.

Although Reviewer 2 recommended reject, I think that the comments could be addressed in a major revision, particularly, the concerns of better summarizing the results of these studies.

We would appreciate receiving your revised manuscript by Oct 28 2019 11:59PM. To enhance the reproducibility of your results, we recommend that if applicable you deposit your laboratory protocols in protocols.io, where a protocol can be assigned its own identifier (DOI) such that it can be cited independently in the future. For instructions see: http://journals.plos.org/plosone/s/submission-guidelines#loc-laboratory-protocols

We look forward to receiving your revised manuscript.

Kind regards,

James C. Nieh, Ph.D.

Academic Editor

PLOS ONE

Journal Requirements:

1. Thank you for including the following funding information within your acknowledgements section of your manuscript; "This research was funded as part of the PROTECTS project by the Department of Agriculture, Food and the Marine’s Competitive Research Funding Programme (Grant Award No.17/S/232)."

"he funders had no role in study design, data collection and analysis, decision to publish, or preparation of the manuscript"

Reviewers' comments:

Reviewer's Responses to Questions

**Comments to the Author**

1. Is the manuscript technically sound, and do the data support the conclusions?

Reviewer #1: Yes

Reviewer #2: Yes

2. Has the statistical analysis been performed appropriately and rigorously? 

Reviewer #1: N/A

Reviewer #2: Yes

3. Have the authors made all data underlying the findings in their manuscript fully available?

Reviewer #1: Yes

Reviewer #2: Yes

4. Is the manuscript presented in an intelligible fashion and written in standard English?

Reviewer #1: Yes

Reviewer #2: Yes

5. Review Comments to the Author

Reviewer #1: Cullen et al. PLOSONE

In the paper entitled “Fungicides, Herbicides and bees: A systematic review of existing research and methods” the authors adopted a systematic approach to review existing research on the potential impacts of fungicides and herbicides on bees. The study is very interesting and topical since many recent studies highlighted the potential direct or indirect impact of fungicides and herbicides on bee health (e.g. Simon-Delso et al. 2014 PlosOne; Motta et al. 2018 PNAS). I found the methods adequate and the results clearly presented.

Minor comments:

L153-157: Maybe you missed the geographical distribution of one study. You have 90 studies but 43+30+8+6+1+1=89. Please, double-check the information reported in this paragraph and in the figure 1;

L213-217: Here, you provide the information on the life stage that the studies focused on but it would be interesting to also know in which stage the bees were exposed to pesticides;

L274-278: Differences in life history traits between honey bees and solitary bees may also determine differences in routes and levels of pesticide exposure. These differences were recently highlighted in a Workshop on ‘Pesticide Exposure Assessment Paradigm for non-Apisbees’ held in 10–12 January 2017, at the United States Environmental Protection Agency (USEPA) in Arlington, Virginia (USA) and published in a special collection (https://academic.oup.com/ee/pages/pesticide_exposure_in_non-honey_bees).

L285: Please, add the percentage of studies that investigated sub-lethal effects;

Table 3 and 4: Add a column with the pesticide group for each compound.

Reviewer #2: In “Fungicides, Herbicides and bees: A systematic review of existing research and methods”, the authors summarize what types of studies have investigated the impact of fungicides and herbicides on bees. Given their findings, the authors argue that further work must be done on non-honey bees as well as sub-lethal effects of fungicides and herbicides (which are generally less studied than neonic pesticides).

Major comment:

Although this review succeeds at illuminating that more work should be done on the effects of fungicides and herbicides on bees, it would be greatly improved by including a summary of the results of the studies. Do fungicides and herbicides affect bees? A review of this nature would add greatly to the knowledge about the effect of fungicides and herbicides on bees by synthesizing the findings of the reviewed studies, rather than only summarizing what methods prior research has employed. In its current state, this review is incomplete. The addition of a synthesis and summary of how fungicides and herbicides affect bees will greatly improve the manuscript and fill many of the gaps of knowledge the authors argue need filling.

Minor comments:

Line 25: *have been

-Line 46 and throughout the entire document: It is unclear when you say “bees” whether you are referring to honey bees specifically or all species of bees. In most cases it seems to be in reference to honey bees, but at other times it is more general.

-Some of the questions outlined in the intro are unclear or hard to understand. Clearer wording would make this more helpful:

-Line 68: I do not understand question (i). Does ‘when’ mean year, season, stage of bee lifecycle? Does ‘where’ mean part of the world, type of crop, etc.?

-Line 71: What is the meaning of “formulation” in this context?

-Line 97: Why did you not include observational data? I would think that observations of declines in bee communities would be important to an understanding of how fungicides and herbicides affect bees. A justification for this omittance should be included.

-Line 117: Forgot to include definition of ‘model’ studies.

-Line 144: Please elucidate what a ‘consumption effect’ is.

-Line 215: Were studies that looked at egg, pupa, and larvae exclusive to honey bee studies, or were these common for other species as well?

6. PLOS authors have the option to publish the peer review history of their article (what does this mean?). If published, this will include your full peer review and any attached files.

Reviewer #1: Yes: Fabio Sgolastra

Reviewer #2: No

---

## [Author Response · Author response to Decision Letter 0]

30 Oct 2019

esponses to Review Comments to the Author

Reviewer #1: Cullen et al. PLOSONE

In the paper entitled “Fungicides, Herbicides and bees: A systematic review of existing research and

methods” the authors adopted a systematic approach to review existing research on the potential

impacts of fungicides and herbicides on bees. The study is very interesting and topical since many

recent studies highlighted the potential direct or indirect impact of fungicides and herbicides on bee

health (e.g. Simon-Delso et al. 2014 PlosOne; Motta et al. 2018 PNAS). I found the methods

adequate and the results clearly presented.

Minor comments:

L153-157: Maybe you missed the geographical distribution of one study. You have 90 studies but

43+30+8+6+1+1=89. Please, double-check the information reported in this paragraph and in the

figure 1;

This has now been checked and rectified.

L213-217: Here, you provide the information on the life stage that the studies focused on but it

would be interesting to also know in which stage the bees were exposed to pesticides;

Here we have already included data no both the life-stage that the bees were exposed to pesticides

and the life-stage at which effects were measured. We have clarified this now in the methods section

as follows: “If the life stages at which bees were exposed to pesticides different to the life stage at

which effects were measured, then both were included in final analyses”.

L274-278: Differences in life history traits between honey bees and solitary bees may also determine

differences in routes and levels of pesticide exposure. These differences were recently highlighted in

a Workshop on ‘Pesticide Exposure Assessment Paradigm for non-Apisbees’ held in 10–12 January

2017, at the United States Environmental Protection Agency (USEPA) in Arlington, Virginia (USA) and

published in a special collection (https://academic.oup.com/ee/pages/pesticide_exposure_in_nonhoney_

bees).

We thank the reviewer for pointing this out, and have now cited two of these papers at this point

within the manuscript (L292).

L285: Please, add the percentage of studies that investigated sub-lethal effects;

This has been added as suggested (L296 & L299)

Table 3 and 4: Add a column with the pesticide group for each compound.

These columns have now been added as suggested

Reviewer #2: In “Fungicides, Herbicides and bees: A systematic review of existing research and

methods”, the authors summarize what types of studies have investigated the impact of fungicides

and herbicides on bees. Given their findings, the authors argue that further work must be done on

non-honey bees as well as sub-lethal effects of fungicides and herbicides (which are generally less

studied than neonic pesticides).

Major comment:

Although this review succeeds at illuminating that more work should be done on the effects of

fungicides and herbicides on bees, it would be greatly improved by including a summary of the

results of the studies. Do fungicides and herbicides affect bees? A review of this nature would add

greatly to the knowledge about the effect of fungicides and herbicides on bees by synthesizing the

findings of the reviewed studies, rather than only summarizing what methods prior research has

employed. In its current state, this review is incomplete. The addition of a synthesis and summary of

how fungicides and herbicides affect bees will greatly improve the manuscript and fill many of the

gaps of knowledge the authors argue need filling.

While we agree that it would be very useful to have a review on the effects of fungicides and

herbicides on bees, we think that that is beyond the scope of the current paper. The best way to do

this is to perform a meta-analysis, using effect sizes from a number of studies to make a broad

conclusions as to whether these compounds have effects or not. However, our review shows that

there is not a lot of studies investigating effects of herbicides and fungicides on bees overall, and

where there are they vary hugely in terms of compounds studied, concentrations used, and bee

species studied. To perform a proper meta-analysis you need a cohort of studies investigating a

similar question so that they can be compared and summarised in a meaningful way. This simply

does not exist in this body of literature yet, and we have clarified this further in the discussion (L387-

399). There are many other existing studies that, like ours, summarise research approaches and

methods rather than effects (e.g. Lundin et al. 2015 Plos One), and this is often the first step towards

a more detailed meta-analysis later on when the existing body of literature has further developed.

Although a full evaluation of effects is not possible, we agree with the reviewer that a higher level

summary of effects would be useful and so we have added details of the main findings of each paper

in S3 Table B in the supplementary materials.

Minor comments:

Line 25: *have been

This has been corrected

-Line 46 and throughout the entire document: It is unclear when you say “bees” whether you are

referring to honey bees specifically or all species of bees. In most cases it seems to be in reference to

honey bees, but at other times it is more general.

In general, wherever we use this term we are referring to all bees. We have clarified this at this point

as suggested here (L47), and also in line 226-230.

-Some of the questions outlined in the intro are unclear or hard to understand. Clearer wording

would make this more helpful:

We have tried to clarify this where possible and have also referred to Table 1 here that provides a

more detailed description of the variables used in each question.

-Line 68: I do not understand question (i). Does ‘when’ mean year, season, stage of bee lifecycle?

Does ‘where’ mean part of the world, type of crop, etc.?

We have clarified this and it now reads “In what year and in what geographical location has existing

research has taken place?”

-Line 71: What is the meaning of “formulation” in this context?

We have clarified this as follows: “commercially produced pesticide formulation”

-Line 97: Why did you not include observational data? I would think that observations of declines in

bee communities would be important to an understanding of how fungicides and herbicides affect

bees. A justification for this omittance should be included.

Observational data is usually based on correlations. Therefore, it is never clear whether any patterns

observed are directly due to pesticide use or perhaps some other intermediary factor. We have now

clarified this further here (L103-104).

-Line 117: Forgot to include definition of ‘model’ studies.

This has now been added

-Line 144: Please elucidate what a ‘consumption effect’ is.

This has now been clarified “consumption (e.g. ability to consume nectar and/or pollen)”

-Line 215: Were studies that looked at egg, pupa, and larvae exclusive to honey bee studies, or were

these common for other species as well?

We have now added in the following text in response to this question: “Egg studies were only carried

out on Megachile rotundata, whereas larval and pupal studies involved multiple bee species

---

## [Decision Letter · Decision Letter 1]

12 Nov 2019

Fungicides, Herbicides and bees: A systematic review of existing research and methods

PONE-D-19-21233R1

Dear Dr. Stanley,

We are pleased to inform you that your manuscript has been judged scientifically suitable for publication and will be formally accepted for publication once it complies with all outstanding technical requirements.

With kind regards,

James C. Nieh, Ph.D.

Academic Editor

PLOS ONE

Additional Editor Comments (optional):

Reviewers' comments:

Reviewer's Responses to Questions

**Comments to the Author**

1. If the authors have adequately addressed your comments raised in a previous round of review and you feel that this manuscript is now acceptable for publication, you may indicate that here to bypass the “Comments to the Author” section, enter your conflict of interest statement in the “Confidential to Editor” section, and submit your "Accept" recommendation.

Reviewer #2: All comments have been addressed

2. Is the manuscript technically sound, and do the data support the conclusions?

Reviewer #2: Yes

3. Has the statistical analysis been performed appropriately and rigorously? 

Reviewer #2: Yes

4. Have the authors made all data underlying the findings in their manuscript fully available?

Reviewer #2: Yes

5. Is the manuscript presented in an intelligible fashion and written in standard English?

Reviewer #2: Yes

6. Review Comments to the Author

Reviewer #2: (No Response)

7. PLOS authors have the option to publish the peer review history of their article (what does this mean?). If published, this will include your full peer review and any attached files.

Reviewer #2: No

---

## [Editor Report · Acceptance letter]

3 Dec 2019

PONE-D-19-21233R1 

Fungicides, Herbicides and bees: A systematic review of existing research and methods 

Dear Dr. Stanley:

I am pleased to inform you that your manuscript has been deemed suitable for publication in PLOS ONE. Congratulations! Your manuscript is now with our production department. 

With kind regards,

on behalf of

Dr. James C. Nieh 

Academic Editor

PLOS ONE